# Multidrug-Resistant *Proteus mirabilis* Strain with Cointegrate Plasmid

**DOI:** 10.3390/microorganisms8111775

**Published:** 2020-11-12

**Authors:** Andrey Shelenkov, Lyudmila Petrova, Valeria Fomina, Mikhail Zamyatin, Yulia Mikhaylova, Vasiliy Akimkin

**Affiliations:** 1Central Research Institute of Epidemiology, Novogireevskaya str. 3a, 111123 Moscow, Russia; mihailova@cmd.su (Y.M.); vgakimkin@yandex.ru (V.A.); 2National Medical and Surgical Center named after N.I. Pirogov, Nizhnyaya Pervomayskaya str., 70, 105203 Moscow, Russia; lutix85@yandex.ru (L.P.); med_2006@mail.ru (V.F.); mnz1@yandex.ru (M.Z.)

**Keywords:** *Proteus mirabilis*, horizontal gene transfer, antibiotic resistance, MDR, CRISPR-Cas system, phage genome

## Abstract

*Proteus mirabilis* is a component of the normal intestinal microflora of humans and animals, but can cause urinary tract infections and even sepsis in hospital settings. In recent years, the number of multidrug-resistant *P. mirabilis* isolates, including the ones producing extended-spectrum β-lactamases (ESBLs), is increasing worldwide. However, the number of investigations dedicated to this species, especially, whole-genome sequencing, is much lower in comparison to the members of the ESKAPE pathogens group. This study presents a detailed analysis of clinical multidrug-resistant ESBL-producing *P. mirabilis* isolate using short- and long-read whole-genome sequencing, which allowed us to reveal possible horizontal gene transfer between *Klebsiella pneumoniae* and *P. mirabilis* plasmids and to locate the CRISPR-Cas system in the genome together with its probable phage targets, as well as multiple virulence genes. We believe that the data presented will contribute to the understanding of antibiotic resistance acquisition and virulence mechanisms for this important pathogen.

## 1. Introduction

*Proteus mirabilis* is a member of the Morganellaceae family of gram-negative bacilli. Before 2016, it had been assigned to Enterobacteriaceae family. Bacteria of the genus *Proteus* possess swarming motility and do not form capsules. They represent anaerobic bacteria found in soil, wastewater environments, and the normal intestinal microflora of humans and animals, but they can also be revealed in hospital settings [1,2]. The ability of *Proteus* organisms to produce urease and to alkalinize the urine by hydrolyzing urea to ammonia makes them effective in producing an environment in which they can survive. This leads to precipitation of organic and inorganic compounds, which leads to struvite stone formation. Among various *Proteus* species, *P. mirabilis* isolates are clinically significant and usually responsible for urinary tract and wound infections. They are the fifth most common cause of nosocomial urinary tract infections and sepsis in hospitalized individuals [3].

The spread of multidrug-resistant (MDR) *P. mirabilis* isolates producing extended-spectrum β-lactamases (ESBLs) is constantly increasing worldwide. For example, to name a few, *P. mirabilis* strains harboring *bla*_CMY2_ were observed in Ireland [4], New Delhi metallo-beta-lactamase-5 (NDM-5)-producing *Proteus* isolate was analyzed in China [5], and NDM-1 positive strains were recently found in Italy [6] and Tunisia [7]. Some cases of NDM-producing *P. mirabilis* have also been revealed in Brazil [8], Austria [9], India [10], and New Zealand [11]. Fursova et al. characterized MDR *P. mirabilis* clinical isolates from Russia carrying various *bla*_CTX-M_, *bla*_TEM_, and *bla*_OXA_ genes [12]. *P. mirabilis* was also found to be the second most prevalent species, after *Escherichia coli*, among ESBL-producing *Enterobacteriaceae* from chicken meat in Singapore [13], and ESBL production was significantly associated with mortality in patients with bacteremia caused by *P. mirabilis* [14]. Several nosocomial infection outbreaks and community-acquired infections in Ethiopia [15] and Nigeria [16] caused by this bacterial species were also reported.

Studies mentioned above demonstrated that most *bla*-resistance determinants have a plasmid localization, and plasmids play a key role in antimicrobial drug-resistance of *P. mirabilis*. Moreover, such plasmids may have a hybrid origin (cointegrate/mosaic), which is important for the spread of multiple antibiotic resistance genes among *Enterobacterales* [17]. Besides the acquired resistance to the β-lactams, *P. mirabilis* is intrinsically resistant to tetracyclines and polymyxins, including colistin [1], which may facilitate the emergence of multidrug-resistant, or even extensively drug-resistant strains complicating the clinical treatment of infections caused by them. Despite these obvious public health threats, the current level of *P. mirabilis* investigations and whole-genome sequencing is significantly inferior to the ones of ESKAPE pathogens. For example, only 254 whole-genome assemblies of this species are available in Genbank (https://www.ncbi.nlm.nih.gov/genome, accessed on 10 July 2020) as opposed to, for example, *Klebsiella pneumoniae* (9103 assemblies) and *Acinetobacter baumannii* (4523 assemblies).

In this study, we applied the second- and third-generation sequencing to characterize the whole-genome of multidrug-resistant *P. mirabilis* isolate obtained from the patient of Moscow medical center. Long-read whole-genome sequencing using MinION sequencing system (Oxford Nanopore Technologies, Oxford, UK) allowed us to separate the chromosomal and plasmid sequences appropriately. The *P. mirabilis* genome of clinical isolate carried MDR determinants located in chromosome and hybrid (cointegrate) plasmid; furthermore, one of the determinants (*bla*_CTX-M-15_) was characterized by dual localization. The origin of the hybrid plasmid is discussed. This isolate also harbored the bacteriophage genome.

## 2. Materials and Methods

### 2.1. Bacterial Strain Isolation

The sample was obtained from the leg skin ulcer of a 51-year old male patient from the department of thoracic and vascular surgery of a multipurpose medical center in Moscow in 2017. This patient was subjected to the planned surgical intervention regarding arterial sclerosis of lower limbs and had not demonstrated any clinical signs of bacterial infection or sepsis. However, he had a weeping incisional wound and was treated with bacteriophages to prevent further complications.

### 2.2. Determination of Antibiotic Susceptibility

The isolate was identified down to a species level by time-of-flight mass spectrometry (MALDI-TOF MS) using the VITEC MS system (bioMerieux, Marcy-l’Étoile, France). The susceptibility was determined by the disc diffusion method using the Mueller-Hinton medium (bioMerieux, Marcy-l’Étoile, France) and disks with antibiotics (BioRad, Marnes-la-Coquette, France), and by the boundary concentration method on VITEK2Compact30 analyzer (bioMerieux, Marcy-l’Étoile, France). The isolate was tested for susceptibility/resistance to the following drugs: amikacin, amoxicillin/clavulanate, ampicillin, aztreonam, cefazoline, cefepime, cefoperazone/sulbactam, cefotaxime, ceftazidime, ciprofloxacin, colistin, fosfomycin, gentamicin, meropenem, netilmicin, nitrofurantoin, and trimethoprim/sulfamethoxazole. The panel of antimicrobial compounds included for testing in this study reflected those agents used for human therapy in Russian Federation. To interpret the results obtained, we used the clinical guidelines “Determination of the susceptibility of microorganisms to antimicrobial drugs”, version 2015-02 (http://www.antibiotic.ru/minzdrav/files/docs/clrec-dsma2015.pdf), which are based on EUCAST 2015.

### 2.3. DNA Isolation, Sequencing, and Genome Assembly

Genomic DNA was isolated with DNeasy Blood and Tissue kit (Qiagen, Hilden, Germany) and used for paired-end library preparation with Nextera™ DNA Sample Prep Kit (Illumina^®^, San Diego, CA, USA) and whole-genome sequencing (WGS) of the sample. WGS was also performed using the Oxford Nanopore MinION sequencing system (Oxford Nanopore Technologies, Oxford, UK). DNA was used to prepare the MinION library with the Rapid Barcoding Sequencing kit SQK-RBK004 (Oxford Nanopore Technologies, Oxford, UK). The amount of initial DNA used for the barcoding kit was 400 ng for the sample. All mixing steps for DNA samples were performed by gently flicking the microfuge tube instead of pipetting. The library was prepared according to the manufacturer’s protocols. The final library was sequenced on a R9 SpotON flow cell. The standard 24 h sequencing protocol was initiated using the MinKNOW software (Oxford Nanopore Technologies, Oxford, UK). The base-calling of the raw MinION data was performed with Guppy version 3.4.4 (Oxford Nanopore Technologies, Oxford, UK). 

Initially, 2,756,671 pairs of short reads of length 251 were obtained. The reads were prepared for assembly by Q15 filtering, quality trimming at the end of the reads, removal of library adapter sequences, removal of A/T stretches, and excluding short reads (less than 60 bp). As a result, 2,747,335 read pairs left upon filtration that constitutes 99.7% of initial reads. Intermediate assembly of short reads resulted in 62 contigs with a length exceeding 500 base pairs. The longest contig had a length of 1,146,335, and N50 was equal to 239,323 bp, respectively; 92.3% of reads were mapped back to this assembly, and median coverage was equal to 245x. 

The total number of long reads obtained was equal to 271,634. The longest read had a length of 258,531 bp, while the median length was 4603. The sequencing reads (2,747,335 pairs of short reads and 271,634 long reads) were subjected to hybrid assembly using both long and short reads with Unicycler version 0.4.8-beta [18]. Of these reads, 222,644 (82%) long reads were aligned to short-read contigs, and thus, contributed to hybrid assembly.

The final assembly of a strain CriePir89 contained three contigs having lengths 4,292,030 (chromosome, G+C content 39.18%), 71,448 (plasmid, G+C content 42.3%) and 59,078 (phage, G+C content 46.72%). Chromosome and plasmid contigs were circular, as determined by Unicycler. Additional long read mapping to the final assembly was performed for verification purposes and had confirmed this determination.

### 2.4. Data Processing

The assembled genome was processed using a custom software pipeline (as used in Reference [19]), including a set of scripts for the seamless integration of various available software tools. The main goal of investigations was to determine the antibiotic resistance in silico, including both known acquired resistance genes and mutations facilitating the development of such a resistance. The parameters useful for epidemiological surveillance, such as plasmid typing, and the presence of virulence factors, were also studied.

We used ResFinder [20] and CARD [21] databases to find the acquired antibiotic-resistance genes, VFDB [22] and literature analysis to check for the presence of virulence genes, and pMLST tool [23] for plasmid detection and typing.

To build the phylogenetic tree representing relations between CriePir89 and other complete *P. mirabilis* genomes from RefSeq database (224 isolates totally), Prokka [24] was used for gene annotation, Roary [25] for obtaining core genome, and RAxML [26] for construction of the maximum likelihood tree itself. Bootstrap values were calculated by *N* = 1000 iterations.

### 2.5. Ethical Statement

Ethical approval was not required as human samples were routinely collected, and patients’ data remained anonymous. The planning conduct and reporting of the study were in line with the Declaration of Helsinki, as revised in 2013.

## 3. Results

### 3.1. Antibiotic Resistance

The final assembly of a strain CriePir89 contained three contigs having lengths 4,292,030 (chromosome), 71,448 (plasmid) and 59,078 (phage), respectively. This annotation was confirmed using the BLAST search in ‘nt’ database on the NCBI website (accessed 26 April 2020). The hybrid assembly was uploaded to NCBI Genbank under the project number PRJNA645205 (accession numbers CP059056 (chromosome), CP059057 (plasmid) and CP059058 (phage)).

The results of phenotypic and genotypic antibiotic resistance profiling are presented in Table 1.

In addition, *catA1* gene conferring chloramphenicol resistance, as well as *mphE* and *msrE* genes involved in erythromycin resistance were revealed in plasmid sequence. However, these drugs were not included in the panel. Also, *tetJ* conferring tetracycline resistance was found in chromosomal sequence, which is not surprising since *P. mirabilis* possesses intrinsic resistance to this antibiotic [1]. Known chromosomal mutations conferring resistance have not been revealed. The complete list of antimicrobial resistance (AMR) genes revealed, including their sequence coordinates, is presented in Appendix A.

### 3.2. Plasmid Structure

The conjugative plasmid was assigned to IncFII replicon type, MOBP relaxase type, and MPF_T mate-pair formation type using Mob-typer tool (https://github.com/jrober84/mob-typer).

Initial analysis using BLAST search in ‘nt’ database (NCBI) has shown that the plasmid could be roughly divided into three parts: 1–27,000 bp, 27,000–68,000, and 68,000–end. The first part possessed high similarity (99%) with various *K. pneumoniae* plasmids (e.g., pA1-3, LC508263.1), the middle part was homologous to *P. mirabilis* plasmids (e.g., pPM64421b, MF150117.1) with the same level of similarity, and the last part was also similar to *K. pneumoniae* and *E. coli* plasmids (in some cases, to the same sequences as for the first part). These parts were divided by two copies of the IS1 insertion sequence. Interestingly, most AMR genes were contained in the first and the last parts, so that the fraction similar to other *P. mirabilis* plasmids carried just a few such genes. The plasmid structure is shown in Figure 1. The insertion sequences were predicted by ISEscan [27].

Since cointegration as a mechanism for the evolution of *P. mirabilis* plasmids has been described recently [28], we investigated this possibility further. Fortunately, earlier we have sequenced 36 *K. pneumoniae* clinical isolates obtained from the same hospital [19], and three of them were collected during the same period as the *P. mirabilis* isolate under investigation. Two of these *K. pneumoniae* isolates, including CriePir75 and CriePir99, were obtained from the same clinical department as CriePir89. The events of horizontal gene transfer in the intensive care unit between *K. pneumoniae* and other members of Enterobacterales have been described recently [29]. However, such events have not been revealed yet in the hospital from which our isolates were received. The comparison of plasmid sequences of these isolates has shown the partial similarity of CriePir89 and CriePir75 *K. pneumoniae* isolates, the sequence of which was also obtained by hybrid short and long read assembly. A brief graphic illustration of the corresponding alignments is shown in Figure 2, while the complete alignments are presented in Appendix A.

It is easy to see that plasmid have identical, or nearly identical (less than 5 mismatches) sequences at the ranges [1; 8226], [11,356; 12,671], [21,959; 27,206], and [64,627; 71,448] (coordinates are based on *P. mirabilis* plasmid sequence). Thus, *aac(3)-IIa, aac(6′)-Ib-cr, bla_CTX-M-15_, bla_OXA-1_*, and *sul1* resistance genes were the same between these two plasmids from two species, so that we may assume the cointegration as a mechanism of CriePir89 plasmid formation.

In addition, *blaTEM-1B, catA1*, and *dfrA1* genes from CriePir75 plasmid were also present in CriePir89, but in chromosome sequence. However, they were also located in the region flanked by two IS1 copies (4,107,926–4,112,492, see Appendix A for details).

### 3.3. Virulence Factors

*P. mirabilis* has several virulence factors, some of which are encoded by operons of virulence genes [30]. CriePir89 encodes *ure* operon, including *ureABCDEFG* genes responsible for urease production. This enzyme contributes to hydrolyzing urea to release ammonia, thus increasing urinary pH, which, in turn, facilitates bacterial adherence and biofilm formation [31]. Another virulence gene is *luxS* involved in quorum sensing. It produces a signal that is important for the interaction of species in the polymicrobial community, which, in turn, plays the key role in virulence gene regulation [32]. 

One of the prominent features of *P. mirabilis* is its swarming ability on solid surfaces, and CriePir89 encodes *cheW* gene contributing to this phenomenon [30]. Additional genes involved in the swarming are the ones from *mr/p* gene cluster encoding fimbria. This cluster in CriePir89 is presented by *mrpA* gene, which also contributes to adherence of bacteria to the epithelial tissue and biofilm formation [33]. Also, *zapA* gene important for regulating *IgA* protease expression at the differentiation of swimmer cells to swarmer cells [34] was revealed.

The hemolytic activity of *P. mirabilis* is related to hemolysin *hpmA* and *hpmB* proteins. *hpmA*, which is encoded by CriePir89, is mainly responsible for tissue damage [35]. Finally, *rpoA* gene associated with urovirulence and antibiotic efficiency [36] was revealed in this strain.

All of the virulence genes reported above were located on the chromosome.

Thus, we can conclude that since CriePir89 possesses large number of virulence factors described for *P. mirabilis*, it has strong dissemination potential.

### 3.4. CRISPR-Cas System

CRISPR arrays were identified in genomic sequence of *P. mirabilis* using CRISPRCasFinder [37]. Four CRISPR candidates were revealed in the chromosome sequence, one of which, located between positions 2,637,474 and 2,638,295 in the chromosome, was given a highest evidence level 4 and contained 13 spacers. Another candidate with four spacers was found in a range [2,628,285; 2,628,551] and was given an evidence level 3, while two others had two spacers each and were assigned to the lowest evidence level 1. In addition, CAS-TypeI-E system, including *cas2*, *cas1*, *cas6*, *cas5*, *cas7*, *cse2*, *cse1*, and *cas3* genes, was found in the vicinity of this CRISPR candidates [2,628,610; 2,637,229]. 

Plasmid and phage sequences also contained CRISPR candidates—two and one, respectively, each of which included two spacers and was assigned an evidence level 1. The locations were [42,589; 42,701] and [42,780; 42,900] for plasmid and [6666; 6836] for phage sequence, respectively. Interestingly, CRISPRTarget tool [38] has revealed several protospacer nucleotide sequences serving as targets for CRISPR RNAs corresponding to Proteus phages in the latter sequence. In addition, such phage protospacer targets were also revealed in a chromosome sequence. This fact complies with the hypothesis that the presence of such phage protospacer target sequences may indicate the exposure of a specific bacterial strain to this or similar phage in the past [38].

### 3.5. Phage Genome

The genome assembly of CriePir89 contained a 59,078 bp contig assigned to a bacteriophage named ASh-2020a by NCBI team. The genome of ASh-2020a is very similar to the one of PM87 (MG030346.1, Novosibirsk, Russia) and P16-2532 (MN840486.1, Moscow, Russia). PM87, isolated from cattle and poultry samples, has demonstrated lytic activity against sensitive strains of *P. mirabilis* [39]. However, PM87 and P16-2532 were more similar to each other than to Ash-2020a (47 vs. ~400 mismatches each, respectively). Currently, these strains are not used for human infection treatment, and the patient was not treated with phages specific to *P. mirabilis*.

### 3.6. Phylogenetic Comparison

Maximum-likelihood phylogenetic core genome-based tree for the closest 11 strains from Genbank is shown in Figure 3. According to phylogenetic analysis, the closest genomic sequence from Genbank belonged to *P. mirabilis* strain 1023322 (GCF_003687785.1, USA, no host data available). The differences based on the core genome built for all strains available in Genbank (containing 1064 genes) for this strain and CriePir89 included 56 mismatches and 210 indels. However, the whole-genome comparison revealed more than 2000 mismatches, and in addition, the genome size of a strain 1023322 was about 10% lower than that of CriePir89, which makes it unlikely that these two strains have the same origin.

Other strains constituting the nearest taxonomic group included SC90 (GCF_009821675.1, China, isolated from chicken), AHEPA923 (GCF_007004575.1, Greece, isolated from human), and NIVEDI3-PG74 (GCF_001640165.1, India, isolated from pig). A large number of core genome mismatches, as well as the variability of isolation sources and countries in this group, suggest that the *P. mirabilis* genomic data available currently do not provide sufficient information for genomic epidemiology investigations. Additional factor confirming this conclusion is a large number of total genes for the whole set of isolates—22,249, while this number for the clade containing CriePir89 is just 5774, which shows high variability among the whole set of available genomes and highlights the need for additional genomic data for developing better typing schemes and increasing the value of phylogenetic investigations.

## 4. Discussion

Although *P. mirabilis* is usually described as an opportunistic pathogen with rather low virulence [40], it is the most commonly isolated species from clinical samples [1]. For example, *P. mirabilis* was responsible for 13.3% of the infections in intensive care units of Brazil in 2011, behind just *K. pneumoniae* [41]. In contrast to ESKAPE pathogens, *P. mirabilis* is not considered as a reservoir of plasmid-encoded AMR genes by some authors (e.g., Reference [42]), but recent reports have demonstrated the possibility of frequent plasmid-mediated AMR gene transfer for this species [43,44]. Moreover, *P. mirabilis* lineage representing a hidden reservoir of OXA-23 and OXA-58 carbapenemases was reported [45]. In addition to natural resistance to colistin, nitrofurans, tigecycline, and tetracycline [46] and increasing resistance to ESBLs [47], this makes *P. mirabilis* a source of emerging healthcare concerns.

CriePir89 contains both chromosomal and plasmid AMR genes. Five genes were found in the plasmid regions that are likely to originate from *K. pneumoniae* plasmid of the isolate obtained from the same hospital department during the same period. The possibility of such cointegration has been reported earlier for *P. mirabilis* [28,48] and other species [49,50], but, to the best of our knowledge, no plasmid gene transfer between *K. pneumoniae* and *P. mirabilis* has been confirmed yet by long-read sequencing. The exact series of events that led to the formation of such a cointegrate plasmid cannot be determined, due to the absence of complete sequences of all evolutionary intermediates that existed between ancestral plasmids of both species. However, three likely parental plasmids were found in the *K. pneumoniae* population simultaneously, one of which was previously sequenced on MinION (Oxford Nanopore Technologies, Oxford, UK). The mechanisms of such cointegration and later development and transfer of such plasmids is yet to be elucidated.

Plasmid AMR genes include *bla_OXA-1_*, which encodes one of the most prevalent narrow-spectrum oxacillinases in *P. mirabilis* [51], β-lactamase gene *bla_CTX-M-15_*, and *aac(6′)-Ib-cr*, which was previously found to be significantly associated with the ESBL genes [51]. The latter gene is likely to be acquired during the horizontal transfer from *K. pneumoniae*, and encodes *aac(6′)-Ib-cr* enzyme that confers ciprofloxacin resistance by its acetylation [52]. The next pair of genes that was not found in related *K. pneumoniae* plasmid consisted of *msrE* and *mphE* (erythromycin and other macrolides). Another cluster of closely located AMR genes included *dfrA12* (trimethoprim), *aadA2* (streptomycin and spectinomycin), and *sul1* (sulphonamides), the latter two carried by TnAs3-like transposon.

Interestingly, chromosomal AMR genes also included *bla_CTX-M-15_* and *sul1*, but also contained several unique genes like *sul2* (sulphonamides) and *bla_TEM-1_* ESBL gene. *bla_TEM-1_* was carried by two copies of class 2 transposon Tn3, which complies with previous findings [53].

CriePir89 was susceptible to meropenem and cefoperazone/sulbactam, which is rather common for clinical *P. mirabilis* isolates [54].

A better understanding of AMR acquisition by *P. mirabilis* will become more straightforward as more complete genomes, especially the ones sequenced using long-read technologies, become available. Currently, the number of such genomes available in public databases is rather low (less than 250 genomes, and just a few sequenced by third-generation sequencers).

Another interesting feature of CriePir89 is the presence of CRISPR-Cas system, which is encoded only in about one-third of sequenced *P. mirabilis* genomes [42]. In this case, CAS-TypeI-E system was detected. The possible phage protospacer targets have been detected in chromosomal sequence, and an integrated phage genome was revealed by hybrid assembly. The encoding of phage genomes by *P. mirabilis* has been reported earlier [55], and it is likely that phages account for some of the observed differences in swarming behavior between strains [42].

In addition, CriePir89 encodes multiple virulence factors, including the genes responsible for urease production, swarming ability, and hemolytic activity. Further investigation of virulence factors may contribute to the filling of knowledge gaps that exist currently—for example, how important is swarming to virulence and why *P. mirabilis* encodes an extensive array of adherence factors, and what are their targets [1].

Virulence factors, antibiotic resistance genes, CRISPR-Cas system, and other isolate characteristics that can be derived from its genomic sequence constitute useful features for the dissemination of pathogenic bacteria and resistance acquisition by them. However, from the epidemiological point of view, some typing schemes based on unique sequence properties or profiles are more pertinent to the isolate classification and surveillance. Such schemes may include well-known multilocus sequence typing, O-antigens of the lipopolysaccharides [56], frequency-based genomic sequence characteristics [57,58], CRISPR sequences [59], or capsule synthesis loci (K-loci) [60]. Although some efforts have been recently made to propose such typing profiles for *P. mirabilis* (e.g., Reference [61]), currently, no generally accepted and reliable scheme exists for this species. Thus, the comprehensive analysis of available *P. mirabilis* genomic sequences, like the one performed by us, may contribute to solving this important problem.

Although the current study is limited to just one isolate, it reflects an interesting fact of horizontal gene transfer as a mechanism of antibiotic resistance acquisition in *P. mirabilis* species. Future investigations may reveal more such transfer events in hospital settings. However, the detailed epidemiological characterization of *P. mirabilis* population lies beyond the scope of the current investigation.

We believe that the detailed characterization of antibiotic resistance and virulence factors of this isolate will facilitate the investigations of multidrug-resistant *P. mirabilis* emergence and spreading—and will ultimately lead to new effective treatment strategies.

## 5. Conclusions

Here we report the MDR and virulent *P. mirabilis* isolate from Moscow, Russia carrying plasmid-mediated resistance to fourth-generation cephalosporins, aminoglycosides, and macrolides, as well as ESBL genes. The genome was assembled using hybrid second- and third-generation sequencing that identifies the possible cointegrate origin of the plasmid containing several AMR genes previously revealed in *K. pneumoniae* plasmid from the same hospital. CAS-TypeI-E system was identified in the genome, and possible phage protospacer targets have been detected, as well as a complete integrated phage genome. We believe that the data obtained will contribute to a better understanding of the AMR acquisition mechanisms and dissemination potential of this important pathogen, which is currently less studied than the members of the ESKAPE group and deserves more attention in future investigations.

## Figures and Tables

**Figure 1 microorganisms-08-01775-f001:**
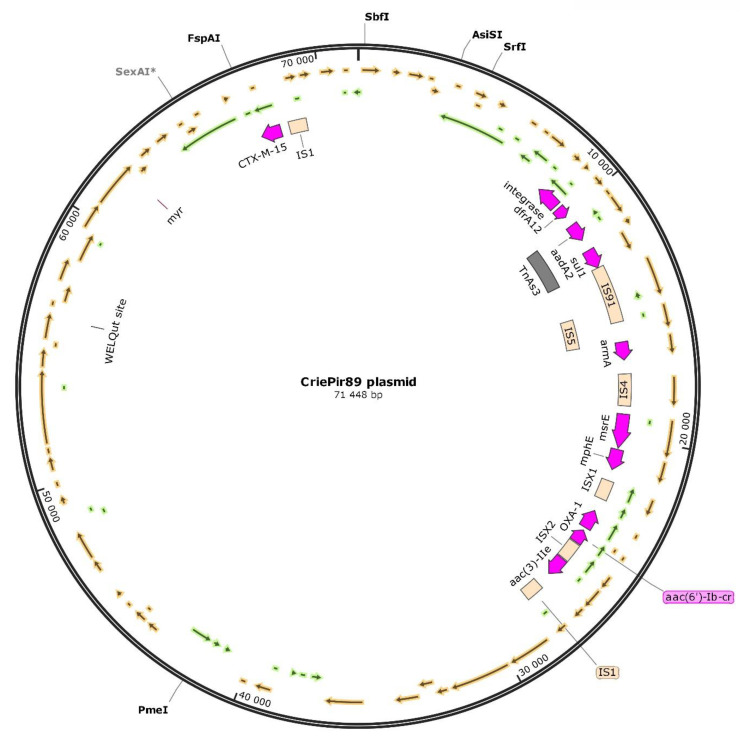
Plasmid structure for CriePir89 isolate.

**Figure 2 microorganisms-08-01775-f002:**
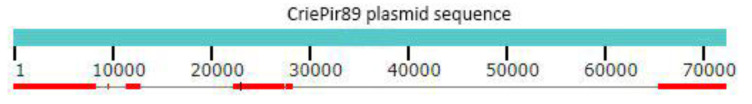
Graphic illustration of CriePir89 plasmid parts identical to the parts of CriePir75 plasmid. Identical parts are shown in red.

**Figure 3 microorganisms-08-01775-f003:**
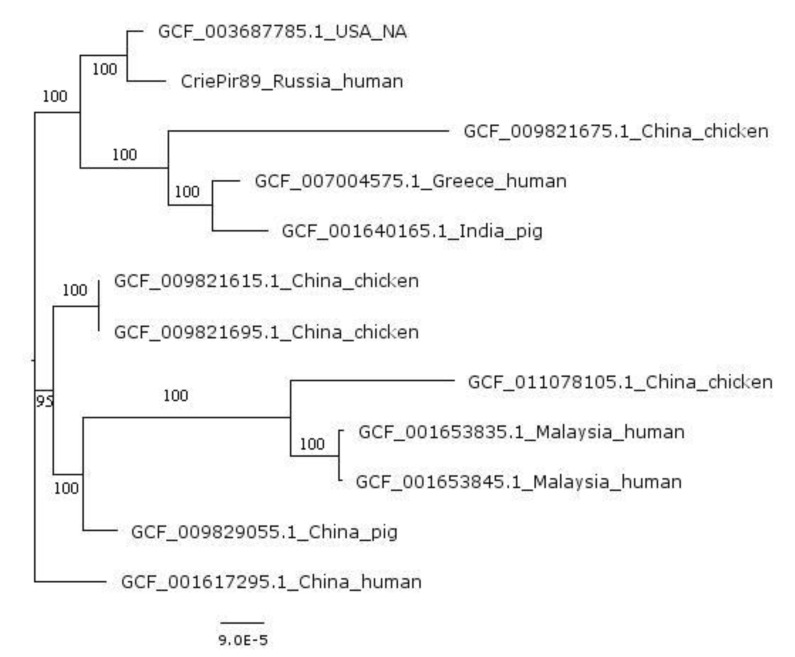
Phylogenetic tree for the nearest neighbors of CriePir89 from Genbank. Country and source of origin are indicated.

**Table 1 microorganisms-08-01775-t001:** Antibiotic resistance of CriePir89 and corresponding resistance genes.

Antibiotic	Result *	MIC	Resistance Genes
amikacin	R	≥64	armA, rmtB
amoxicillin/clavulanate	R	≥32	blaCTX-M-15, blaOXA-1, blaTEM-1B
ampicillin	R	≥32	blaCTX-M-15, blaOXA-1, blaTEM-1B
aztreonam	R	4	blaCTX-M-15
cefazoline	R	≥64	blaCTX-M-15
cefepime	R	16	blaCTX-M-15, blaOXA-1
cefoperazone/sulbactam	S	≤8	-
cefotaxime	R	≥64	blaCTX-M-15
ceftazidime	R	4	blaCTX-M-15
ciprofloxacin	R	≥4	aac(6′)-Ib-cr, qepA1
colistin	R	≥16	intrinsic
fosfomycin	S	≤16	-
gentamicin	R	≥16	aac(3)-IIa, aac(3)-IId, armA, rmtB
meropenem	S	≤0.25	-
netilmicin	R	≥32	aac(3)-IId, aph(3′)-Ia
nitrofurantoin	R	256	intrinsic
trimethoprim/sulfamethoxazole	R	≥320	dfrA1, dfrA12, dfrA17, sul1, sul2

* ‘R’—resistant, ‘S’—susceptible.

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
