# Peer review of "Multidrug-Resistant *Proteus mirabilis* Strain with Cointegrate Plasmid"

_microorganisms, 2020, doi:10.3390/microorganisms8111775_

Round 1
Reviewer 1 Report
Shelenkov et al. provide a detailed analysis of clinical multidrug-resistant ESBL-producing P. mirabilis isolate using short- and long-read whole genome sequencing.
However, the study is restricted to only one sample and this is too small a number. Ideally, the study should be extended to include more samples, which should be sequneced.
In addition, I have the following comments,
1.The authors state that they have performed 2nd and 3rd generation sequencing. However, very little detail about Illumina sequencing is provided.
2. No statistics details about the assembly is provided.
3. It would be good to have a Bandage figure of the three contigs.
4. Line 157-158: Authors should discuss about HGT in the same clinical settings.
5. The discuss is too small. Please discuss your results extensively in detail linking your findings to the current literature.
Author Response
We would like to thank the reviewer for useful suggestions that has led to significant improvements in our manuscript readability. Our comments are given below boldfaced.
Shelenkov et al. provide a detailed analysis of clinical multidrug-resistant ESBL-producing P. mirabilis isolate using short- and long-read whole genome sequencing.
However, the study is restricted to only one sample and this is too small a number. Ideally, the study should be extended to include more samples, which should be sequneced.
We agree that sequencing multiple P. mirabilis isolates would make the study more interesting. However, current manuscript is devoted not to epidemiological surveillance or other bacterial population investigations of P. mirabilis, but rather to reporting the event of horizontal gene transfer between K. pneumoniae and single P. mirabilis isolate. Such a transfer has not been revealed for other available P. mirabilis isolates, so we have not included them in the current study.
In addition, I have the following comments,
1.The authors state that they have performed 2nd and 3rd generation sequencing. However, very little detail about Illumina sequencing is provided.
Illumina sequencing details were provided in the Section 2.3 (DNA isolation, sequencing and genome assembly), at the end of the section.
- No statistics details about the assembly is provided.
Statistics was added to the Section 2.3 (DNA isolation, sequencing and genome assembly), at the end of the section.
- It would be good to have a Bandage figure of the three contigs.
Thank you for the suggestion. However, we believe that Bangage contig representation is not appropriate in our case, since our contigs assembled using MinION data include just single contigs for the chromosome, plasmid and phage genome, respectively, thus additional assembly or linkage is not required. Illumina sequencing statistics was provided as suggested, but we think that since short-read assembly, for analysis of which the Bandage was initially developed, does not represent the final assembly, but rather forms just an intermediate point of this process, its description would be redundant and would not provide additional insights into genome structure.
- Line 157-158: Authors should discuss about HGT in the same clinical settings.
Additional discussion and references were added.
- The discuss is too small. Please discuss your results extensively in detail linking your findings to the current literature.
Discussion section has been extended according to the suggestions provided.
Reviewer 2 Report
The manuscript "Multi-drug Resistant Proteus mirabilis Strain with Cointegrate Plasmid" is well written and provides covers aim outlined by the authors. The manuscript mentions several genomic techniques to characterize the antibiotic/drug resistance in P. mirabilis.
Kindly mention the bootstrap values on the tree nodes. There are few grammatical errors. Other than that, the manuscript is in decent form.
Author Response
We would like to thank the reviewer for useful suggestions that has led to improvements in our manuscript readability. Our comments are given below boldfaced.
The manuscript "Multi-drug Resistant Proteus mirabilis Strain with Cointegrate Plasmid" is well written and provides covers aim outlined by the authors. The manuscript mentions several genomic techniques to characterize the antibiotic/drug resistance in P. mirabilis.
Kindly mention the bootstrap values on the tree nodes. There are few grammatical errors. Other than that, the manuscript is in decent form.
Bootstrap values have been added to the tree (Fig. 3), and the tree description has been updated in the text (the end of section 2.4)
Reviewer 3 Report
The article addresses a major problem in the field of microbiology
Introductions
Page 1
Line 27 please delete space before new sentence starting with Bacteria
Line 34-35 "They are the fifth most common cause of nosocomial urinary tract infections and sepsis in hospitalized individuals". The reference than you used "[1], Jessica N. Schaffer et al." is review article and do not support your article. It is better to cite one article which support the most common cause of nosocomial urinary tract in your country or region.
Line 37 “to name just a few” is better to change to “to name but a few”, that is a typical English phase
Line 45 please delete space before new sentence starting with Several
Page 2
Line 48 please change “The studies mentioned above have demonstrated” to “Studies mentioned above demonstrated”
Line 64 please delete space before new sentence starting with The
Lines 68-70 “We believe that detailed characterization of antibiotic resistance and virulence factors of this isolate will facilitate the investigations of multidrug-resistant P. mirabilis emergence and spreading and will ultimately lead to development of new effective treatment strategies” This paragraph could be redirected to discussion section .
Materials and Methods
The session "2.1. DNA isolation, sequencing, and genome assembly" must rewritten, because you firstly test you isolates for antibiotic susceptibility, and if this strain is resistant you proceed your study at gene level.
I suggest:
"2.1 Bacterial strain isolation" In this session you describe your patient, line 73-77 in your article.
2.2 "Determination of antibiotic susceptibility"
2.3 DNA isolation, sequencing, and genome assembly
Page 8
Discussion
Line 243 The first sentence is too long.
Please add full stop after reference 38.
Lines 247-249 Please add some verb in sentence "In addition to natural resistance to colistin, nitrofurans, tigecycline and tetracycline [43] and increasing resistance to ESBLs [44], this makes P. mirabilis a source of emerging healthcare concerns". Moreover, after "in addition" must add comma.
Please add paragraph with limitations of your study.
Author Response
We would like to thank the reviewer for useful suggestions that has led to significant improvements in our manuscript readability. Our comments are given below boldfaced.
The article addresses a major problem in the field of microbiology
Introductions
Page 1
Line 27 please delete space before new sentence starting with Bacteria
Fixed.
Line 34-35 "They are the fifth most common cause of nosocomial urinary tract infections and sepsis in hospitalized individuals". The reference than you used "[1], Jessica N. Schaffer et al." is review article and do not support your article. It is better to cite one article which support the most common cause of nosocomial urinary tract in your country or region.
Reference updated.
Line 37 “to name just a few” is better to change to “to name but a few”, that is a typical English phase
Changed as suggested.
Line 45 please delete space before new sentence starting with Several
Fixed.
Page 2
Line 48 please change “The studies mentioned above have demonstrated” to “Studies mentioned above demonstrated”
Changed as suggested.
Line 64 please delete space before new sentence starting with The
Fixed.
Lines 68-70 “We believe that detailed characterization of antibiotic resistance and virulence factors of this isolate will facilitate the investigations of multidrug-resistant P. mirabilis emergence and spreading and will ultimately lead to development of new effective treatment strategies” This paragraph could be redirected to discussion section .
The paragraph has been moved to Discussion
Materials and Methods
The session "2.1. DNA isolation, sequencing, and genome assembly" must rewritten, because you firstly test you isolates for antibiotic susceptibility, and if this strain is resistant you proceed your study at gene level.
I suggest:
"2.1 Bacterial strain isolation" In this session you describe your patient, line 73-77 in your article.
2.2 "Determination of antibiotic susceptibility"
2.3 DNA isolation, sequencing, and genome assembly
Revised as suggested.
Page 8
Discussion
Line 243 The first sentence is too long.
Please add full stop after reference 38.
Revised as suggested.
Lines 247-249 Please add some verb in sentence "In addition to natural resistance to colistin, nitrofurans, tigecycline and tetracycline [43] and increasing resistance to ESBLs [44], this makes P. mirabilis a source of emerging healthcare concerns". Moreover, after "in addition" must add comma.
The verb here is ‘makes’, and the part of the sentence starting with ‘in addition’ ends right before ‘this makes’, so comma is located there.
Please add paragraph with limitations of your study.
The paragraph has been added:
Although the current study is limited to just one isolate, it reflects an interesting fact of horizontal gene transfer as a mechanism of antibiotic resistance acquisition in P. mirabilis species. The future investigations may allow to reveal more such transfer events in hospital settings. However, detailed epidemiological characterization of P. mirabilis population lies beyond the scope of the current investigation.
Round 2
Reviewer 1 Report
All my comments have been addressed. However, I would like to see Bandage figures.